# Towards the Configuration of a Photoelectrocatalytic Reactor: Part 2—Selecting Photoreactor Flow Configuration and Operating Variables by a Numerical Approach

**DOI:** 10.3390/nano12173030

**Published:** 2022-08-31

**Authors:** Daniel Borrás-Jiménez, Wilber Silva-López, César Nieto-Londoño

**Affiliations:** 1Grupo de Investigación en Óptica y Espectroscopía, Universidad Pontificia Bolivariana, Medellín 050031, Colombia; 2Grupo de Investigación en Energía y Termodinámica, Universidad Pontificia Bolivariana, Medellín 050031, Colombia

**Keywords:** photoelectrocatalytic reactor design, photocatalytic space-time yield, Computational Fluid Dynamics, mass transfer, textile dye degradation

## Abstract

This work aims to select a photoreactor flow configuration and operational conditions that maximize the Photocatalytic Space-time Yield in a photoelectrocatalytic reactor to degrade Reactive Red 239 textile dye. A numerical study by Computational Fluid Dynamics (CFD) was carried out to model the phenomena of momentum and species transport and surface reaction kinetics. The photoreactor flow configuration was selected between axial (AF) and tangential (TF) inlet and outlet flow, and it was found that the TF configuration generated a higher Space-time Yield (STY) than the AF geometry in both laminar and turbulent regimes due to the formation of a helical movement of the fluid, which generates velocity in the circumferential and axial directions. In contrast, the AF geometry generates a purely axial flow. In addition, to maximize the Photocatalytic Space-time Yield (PSTY), it is necessary to use solar radiation as an external radiation source when the flow is turbulent. In conclusion, the PSTY can be maximized up to a value of 45 g/day-kW at an inlet velocity of 0.2 m/s (inlet Reynolds of 2830), solar radiation for external illumination, and internal illumination by UV-LEDs of 14 W/m^2^, using a photoreactor based on tangent inlet and outlet flow.

## 1. Introduction

Water covers approximately 71% of the Earth’s surface, with only 2.5% fresh water and 0.007% suitable for human consumption. Its sustainable use is considered according to the “contextual availability”, which depends on incorporating elements such as the ecosystem requirements, human consumption, and anthropic activities [1]. The low availability of water for human consumption, and the high growth in the demand for water resources, have caused some concern in sustainable economic development and have made wastewater treatment and reuse a worldwide commitment. In 2015, the United Nations General Assembly (UN) adopted the 2030 Program for Sustainable Development and, together with world leaders, proposed 17 objectives, among which the 6th goal refers to water and sanitation and imposes goals in terms of the sustainable use of water, wastewater treatment, disposal of discharges and technologies for the water reuse [2].

Synthetic dyes have been the focus of research in environmental remediation due to their high demand and persistence in wastewater; they also cause a decrease in dissolved oxygen in the water, altering the biological activity in aquatic life [3], and have high toxicity, carcinogenicity, and mutagenicity [4,5,6]. To reduce the impact of dyes on the environment, it is necessary to achieve their complete mineralization or the formation of less toxic compounds to increase their biodegradability. Bioremediation technologies such as Sequential Batch Reactor (SBR), Anaerobic-Fluidized Bed Reactor (FBR), and Activated Sludge Process (ASP) have presented significant challenges in the degradation of textile waters due to the non-biodegradability of synthetic dyes, which difficult the microbial growth and process efficiency. In addition, these biological systems require treatment of at least 12 to 24 h, need a large surface area, and produce large amounts of toxic sludge [7]. Advanced Oxidation Processes (AOPs) stand out in environmental remediation due to their efficiency in degrading non-biodegradable recalcitrant pollutants produced in different industrial sectors, a fundamental technology to achieve Sustainable Development Goals and reuse of wastewater. Photoelectrocatalysis is an AOP that is efficient in the mineralization of recalcitrant synthetic dyes without the generation of sludge [8], which leads to being a promising tertiary treatment in industries such as textiles, an industrial sector with the most significant dye discharge into the environment (54%) [9].

Research on photocatalytic reactors has increased exponentially in the last 40 years [10] with a growing up in three fundamental aspects: (1) the light source, (2) the state of the catalyst, and (3) the type of operation. The light source is mainly characterized by its wavelength and the source type. The wavelength range is strongly related to the catalyst type, and its absorption spectrum [8]. The catalyst may be in suspension or supported/immobilized on a substrate. Suspended photocatalysis systems have shown several problems that affect large-scale development, such as the additional energy applied to keep the suspension stable, a post-treatment to separate the photocatalyst particles from water, and kinetic problems related to illumination efficiency [11]. A system based on the immobilized catalyst was developed as a solution. Sundar et al. [12] compared the apparent reaction rate constant, Photocatalytic Space-time Yield, specific removal rate, and electrical power consumption of 24 different types of photocatalytic reactors, founding that the reactors with immobilized catalyst perform better than slurry reactors in environmental remediation applications. In both kinds of systems, the formation of electron $e^{-}$—hole ($h^{+}$) pairs in the photocatalyst are the critical step for excellent catalytic efficiency and the high recombination of photogenerated $e^{-}/h^{+}$ species is a common problem [13]. Therefore, getting an excellent catalytic system implies obtaining a low recombination rate, which is reached when a photocatalytic material is joined with an electrode (supported catalyst) in an electrochemical cell. In that cases, an external polarization potential is applied between two electrodes that produce rapid charge separation on the photocatalyst, reducing the recombination process [14]. This system is called photoelectrocatalysis, or electrochemically-assisted photocatalysis [8]. Papagiannis et al. [15] studied the degradation of the azo dye Basic Blue 41 and found that the degradation by photoelectrocatalysis was 12% higher than by photocatalysis due to the lower recombination of the photogenerated charges.

Many semiconductors with potential uses in photoelectrochemical cells as a photocatalyst, including TiO_2_, ZnO, W_2_O_3_ [13] and others. However, its applications are limited by the OH radical production kinetic, requiring a high production of radicals per second and, therefore, low $e^{-}/h^{+}$ rate recombination. Although they have acceptable properties for photocatalysis, developing new materials is necessary to maximize the OH radicals produced. Xiang et al. [16] experimentally determined a hydroxyl radical production rate of several photocatalysts, the TiO_2_ Degussa P25 and TiO_2_ anatase phase the most efficient. Besides, TiO_2_ presents excellent quantum yield, high capacity for oxidation resistance, long-term stability, low preparation cost, and low toxicity [17].

On the other hand, a new family of nanomaterials based on semiconductors has been established and used for photoelectrocatalysis systems. Compared to the conventional materials, nanostructured TiO_2_ such as tubes, wires, fibres, and dots, among others, exhibits high photoelectric efficiency and photocatalytic activity for photodecomposition due to their high surface area and changes in their electronic structure [18]. A tubular TiO_2_ nanostructure can be synthesized by titanium anodizing, and parameters such as nanotube length and diameter can be controlled to improve photocatalytic properties. Ferraz et al. [19] studied the degradation of azo textile dyes such as Dispersed Red 1, Dispersed Red 13, and Dispersed Orange 1, using TiO_2_ nanotubes in a titanium carrier, demonstrated the effectiveness of photoelectrode (Ti/TiO_2_) and photoelectrocatalysis process for degradation of these dyes, achieving a reduction in total organic carbon (TOC) greater than 87%, therefore a decrease in mutagenic and cytotoxic activity in final waters.

An analysis of reactors based on photocatalysis was performed by Sundar et al. [12], founding that they should be designed taking into account three essential aspects: (i) energy efficiency (reduce unabsorbed photon flux), (ii) mass transfer efficiency, and (iii) high catalyst area in low process volume. They analyze 24 photocatalytic reactors; among these is the Spinning Disc Photocatalytic Reactor, which has shown a high electrical energy consumption compared to the rest of the photoreactors evaluated, and it is not represented in the increase of Space-time Yield (STY) and Photocatalytic Space-time Yield (PSTY). They also mentioned that an increase in STY and PSTY could be obtained with a Plug Flow Photocatalytic Reactor (PFR). This analysis has also been confirmed by other authors [20] and is due to the increase in the mass transfer rate to the photocatalyst. Likewise, a high PSTY can be achieved by implementing LEDs since they increase the illumination efficiency while maintaining low energy consumption.

Geometry is crucial in reactors design; in the scientific literature, photoelectrocatalytic reactors with different shapes have been used to evaluate their performance and their general behavior, the most common are planar geometries [21,22,23,24] and tubular geometries [25,26,27], and can be classified hydrodynamically as PFR, perfectly mixed (CSTR), or particular hydrodynamics. Hydrodynamic profiles have been shown to influence the residence times distribution and yield. Moreira et al. [28] evaluated the effect of the inlet flow configuration on the residence time distribution and the ability to photochemically degrade 3-amino-5-methylisoxazole in an annular photoreactor, being a configuration tangent or parallel to the axial axis of the reactor. In the photoreactor with tangential inlet and outlet, a helical flow was present around the inner tube of the light source, which increased the residence time of the particles inside the reactor and improved mass transport. Other strategies have been shown to generate turbulence and increase mixing levels by modifying the internal configuration inside the photoreactor. Montenegro-Ayo et al. [29] used a continuous PEC reactor to degrade acetaminophen, which integrates an anode-cathode configuration in the form of baffles to redirect the flow and create levels of turbulence, increasing the global mass transfer rate. A similar setup was used by Rezaei et al. [30] for the degradation of phenol by supported photocatalysis. Tedesco et al. [31] hydraulically designed a three-dimensional photoanode honeycomb photochemical/photoelectrochemical reactor to improve mixing levels and mass transfer rate within the reactor.

Photocatalytic processes as tertiary treatment for wastewater remediation are still in a “Technological Research” phase [11]. The efficiency of PEC in dye mineralization has already been demonstrated [19,32], and several simplified reaction mechanisms and empirical kinetics have been verified [33]. Taking into account the photoelectrocatalytic reactors review carried out by McMichael et al. [34], further research on photoelectrode geometry optimization, photoreactor design, photoreactor operation, and modeling is needed to improve the photoelectrocatalysis process and advance the technology on a larger scale.

This paper aims to develop a conceptual design proposal for a photoelectrocatalytic reactor to degrade the Reactive Red 239 textile dye, focused on selecting the photoreactor flow configuration and operating variables. Computational Fluid Dynamics (CFD) simulation was used to describe the phenomena of momentum, species transport, and reaction kinetics through two modeling approaches, Real Geometry-Based (RGB) and Porous Medium (PM), and mathematical modeling based on the Beer-Lambert law for radiation transport. The effect of two photoreactor flow configurations using the RGB approach, with axial and tangential flow inlet, was studied on the STY. Then, a mass transfer and kinetic analysis were performed to maximize the PSTY, which has been used by Leblebici et al. [10] to compare 12 types of photocatalytic reactors. Finally, the length of the photoreactor was determined based on the hydrodynamic profiles using the PM approach. This work pretends to establish a basis for designing photoelectrocatalytic reactors for dye degradation.

## 2. Materials and Methods

The design of a photoelectrocatalytic reactor to degrade dyes consists mainly of determining the ideal photoelectrode geometry and its position within the reactor through the optical thickness (distance between the photoelectrode and the surface where the electromagnetic radiation enters). It is also necessary to define a reactor flow configuration that improves mass transfer and maximizes the illumination efficiency, the photoreactor volume, and the appropriate operating variables to maximize the PSTY.

Figure 1 shows the general procedure for designing the photoreactor. The design concept, the photoelectrode geometry selection, and the optical thickness determination were carried out in a previous work [35]. Two kinds of photoelectrode were used, woven and expanded mesh electrode, and the influence of each geometry, its dimensions, and optical thickness in the mass transfer rate and Radiation Intensity Losses (RIL) was evaluated. It was concluded that a woven mesh electrode geometry and an optical thickness of 1 cm allow for high mass transfer rates and radiation losses below 15%.

This section presents the procedure used to determine photoreactor flow configuration, volume, and operating conditions as shown in Figure 2. The following three activities were carried out,
i.Selection of the photoreactor flow configuration: two configurations were evaluated with axial and tangential flow inlet, and the space-time yield (STY) was determined at different cathode positions in both laminar and turbulent regimes, ii.Determination of operating variables of inlet velocity and energy consumption: the photocatalytic space-time yield (PSTY) was evaluated, which allows maximizing the space-time yield without significantly increasing energy consumption,  iii.Photoreactor length estimation: the length of the reactor was increased, and the velocity profiles were evaluated.

First, the governing equations are described, then the boundary conditions, explaining the numerical solution method and simulation strategy. Finally, the computational domains used for the numerical study are presented.

It is worth mentioning that two modeling approaches were used for the numerical study in this work. The first one is the Real Geometry-Based approach (RGB), in which the exact digital model of the photoelectrode geometry is used. This approach is implemented in selecting the photoreactor flow configuration and operating variables. Furthermore, the second one is the Porous Media approach (PM), in which a porous domain is used in the photoelectrode zone to simplify the CFD model and reduce the computational cost. This last approach is recommended in studies where the geometry is complex, which leads to increasing the number of mesh elements and the computational cost [36]. The PM approach is used in the last activity because increasing the reactor length augments the computational cost if the RGB approach is used; therefore, the PM approach is ideal for carrying out this activity.

### 2.1. Governing Equations

The numerical study was developed through Computational Fluid Dynamics simulations. This section shows the governing equations of momentum transport for the PM modeling approach, the equation used to calculate the photoreactor power consumption, and the numerical solution method. Detailed information about the RGB modeling approach, momentum and species transport models equations, surface kinetics, and boundary conditions can be consulted in the previous work [35].

#### 2.1.1. Momentum and Species Transport Model

The photoelectrocatalytic reactor is studied under different inlet flow conditions, leading to the reactor’s operating in laminar and turbulent regimen conditions.

The Navier-Stokes equations and the Standard κ−ε Reynolds-averaged Navier-Stokes equation (RANS) with constant properties describe momentum transport in the laminar and turbulent regime cases, respectively. A convection-diffusion model is assumed to model the dye transport in the laminar and turbulent regime; the mass diffusivity of the dye in the laminar regime is determined with a theoretical equation used by [33] and in the turbulent regime with the Kays-Crawford model. The enhanced wall treatment models the conservation of momentum and species near the walls in the turbulent regime.

Photocatalyst film’s surface reactions in a photoelectrocatalytic system are complex electrochemical and homogeneous reactions. However, the global reaction can be simplified using empirical equations, accounting for the contribution of the most significant variables, such as the surface intensity of radiation, the concentration of the dye, and the voltage effect. In the RGB modeling approach an empirical surface kinetic reaction was used, which was experimentally obtained over titanium dioxide nanotube in a photoelectrocatalytic microreactor by [33] under conditions that ensured the analysis was performed with no significant mass transfer effects, and the surface kinetics completely limited the overall reaction. If the reader is interested in detailed information on the models used, in [35] it can be found information on the momentum, species transport model and surface kinetic reaction equation in Sections 2.1.1, 2.1.2 and 2.1.5, respectively.

The PM modeling approach solved a porous media model in the photoelectrode zone. In the laminar regime, a source term is added that depends on the permeability (α) and an inertial resistance factor ($C_{2}$) or the porous media. Likewise, the medium’s porosity term (γ) is added to the governing equation’s diffusion, convection, and pressure terms. The momentum transport model is as follows,
(1)$$\rho \gamma \left(\upsilon \;\cdot \;▿\right)\upsilon =▿\;\cdot \;\gamma \left[-P+\mu \left(▿\;\cdot \;\upsilon \right)\right]+\left(\frac{\mu }{\alpha }+\frac{C_{2}\rho }{2}\left|\upsilon \right|\right)\upsilon ,$$

Similarly, in the turbulent regime, the porosity term is added to the Reynolds-averaged Navier-Stokes equations (RANS), and in the turbulent kinetic energy and the dissipation rate of turbulent kinetic energy equations, the momentum transport equation takes the form,
(2)$$\rho \gamma \left(\bar{\upsilon }\;\cdot \;▿\rho \bar{\upsilon }\right)=-▿\gamma \bar{P}+▿\;\cdot \;\gamma \left[\mu \left(▿\bar{\upsilon }+▿{\bar{\upsilon }}^{\left(t\right)}\right)+\tau^{T}\right]+\left(\frac{\mu }{\alpha }+\frac{C_{2}\rho }{2}\left|\upsilon \right|\right)\upsilon$$

It is important to mention that this approach was used only in the photoelectrode zone and for the momentum transport model.

#### 2.1.2. Radiation Model

A simplified one-dimensional radiation model is used to determine the energy consumption in the photoelectrocatalytic reactor. First, the RIL due to the water-dye solution was calculated using a model based on the Beer-Lambert law, as follows,
(3)$$RIL^{UV}=1-e^{aC_{dye}\delta },$$
where *a* is an experimentally determined constant, and takes a value of $-1.72\times 10^{-2}\frac{L}{\mathrm{mg}\;\cdot \;\mathrm{cm}}$ for Ultraviolet-A (UVA) average spectrum (320–400 nm) and $-1.33\times 10^{-2}\frac{L}{\mathrm{mg}\;\cdot \;\mathrm{cm}}$ for a wavelength of 365 nm, δ is the optical thickness in cm (can be internal optical thickness—$\delta_{int}$, and external optical thickness—$\delta_{ext}$) and $C_{dye}$ is the RR239 concentration in mg/L.

For the experimental procedure, the transmittance spectrum (%T) for seven water-RR239 dye solutions, with a concentration between 5 and 240 mg/L, were measured initially by UV-Vis spectroscopy (Termo Scientific Genesys 6, 1 cm cell); then, based on the %T values, the RIL for each water-dye concentrations is estimated (1−%T); and finally, using the Beer-Lambert equation and the RIL obtained, an expression for the average extinction coefficient in UVA is determined, which depends on the constant *a* and the concentration of the dye, as shown in Equation (Equation 3). For more information on the experimental procedure and kinetic models used to determine this coefficient, see Section 2.1.5 in [35].

Then, the energy consumption of the illumination system ($P_{light}$) necessary to achieve a given radiation intensity on the photoelectrode surface was calculated, as follows,
(4)$$P_{light}=\frac{I_{UV}A_{sup,int}}{E_{\mathrm{LED}}}\left(1+RIL_{int}\right)+\frac{I_{UV}A_{sup,ext}}{E_{\mathrm{LED}}}\left(1+RIL_{ext}\right),$$
with $I_{UV}$ as radiation intensity on the photoelectrode surface, $E_{\mathrm{LED}}$ as LED energy efficiency (0.6), $A_{sup,int}$ and $A_{sup,ext}$ as the surface area of the internal illumination tube and external tube, respectively, and $RIL_{int}$ and $RIL_{ext}$ as the losses with respect to the internal and external tube, respectively (calculated with Equation (Equation 3)). The first and second terms in Equation (Equation 4) refer to the losses from the internal and external surface, respectively.

### 2.2. Numerical Solution Method

The governing equations for the momentum and species transport were solved through the finite volume method implemented in the commercial CFD software ANSYS Fluent^®^. The SIMPLE and COUPLED solver was used in the RGB and PM approach, respectively, and second-order Upwind discretization schemes were used in both. Simulation convergence was achieved when residuals were lesser than $1\times 10^{-5}$ for each of the transport properties (Volume-weighted average mass imbalance less than $1\times 10^{-12}$ in the fluid domain), and the standard deviation of the mass fraction at the outlet was stable. Additionally, in the PM approach, it was also monitored that the inlet pressure standard deviation was steady. Finally, the $y^{+}$ value was observed throughout the iterative process in the turbulent regime. This value is set to be approximately 1.5 or lesser, ensuring the accuracy of the wall treatment. In Section 3.1.1 independence, mesh, and converge studies are presented to assess the numerical solution.

### 2.3. Computational Domains

#### 2.3.1. Selection of the Photoreactor Flow Configuration

Two flow configuration geometries are evaluated, with axial and tangential flow inlet, as shown in Figure 3. For each geometry, the distance between electrodes (ψ) was varied in 0.75, 1.00, and 1.25 cm. As a simulation strategy, a periodic rotational domain was used (88.8° of all tubular geometry) as shown in Figure 4. The same photoelectrode area was used in each photoreactor domain to be able to compare the results.

The effect of photoreactor flow configuration and electrode spacing on Space-time Yield in both laminar and turbulent regimes is evaluated by the RGB modeling approach. A mass fraction of RR239 equal to 0 on the photoelectrode surface is considered; this refers to the maximum STY under the conditions studied (inlet velocity of 0.03 and 0.4 m/s for the laminar and turbulent regime, respectively).

The STY relates the amount of dye mass degraded in a specific time and volume of reactor [37], and is calculated as follows
(5)$$STY=\frac{m}{V_{reactor}t}=\frac{\dot{m}}{V_{reactor}}.$$
where $\dot{m}$ is the mass flux of degraded dye in a specific photoreactor volume ($V_{reactor}$). To determine $\dot{m}$ in a more accurate way, the dye flux towards the photoelectrode (degraded amount) was determined using a User Defined Function (UDF). The UDF considers the mass fraction profile in the vicinity of the photoelectrode and solves the diffusion flux equation for the laminar and turbulent regime.

The general procedure used for the convergence study is shown in Figure 5. First, a three-dimensional geometry model is created following the specific photoreactor flow configuration and ψ. Then, unstructured computational meshing is defined with polyhedral elements by controlling the element’s size in the walls. Subsequently, the solution of the governing equations (see Section 2.1) is carried out. The Space-time Yield is calculated considering the simulation results using a UDF as mentioned above. Once STY is calculated, the computational mesh was refined by modifying the wall element size until the difference in STY is less than 10% compared to the STY obtained with the previous mesh size. Finally, the calculation of the STY with an infinite mesh was made using Richardson’s extrapolation, considering the methodology reported in [38]. This procedure was done for the case of ψ equal to 0.75 cm in both geometries, in the laminar and turbulent regime. The computational mesh size with the best results was selected for further studies.

#### 2.3.2. Operating Variables

An analysis of mass transfer, chemical kinetics, and energy consumption was carried out to determine the operating variables using the selected photoreactor flow configuration. The Photocatalytic Space-time Yield was maximized by the RGB approach varying the inlet velocity and the surface radiation intensity. A constant photoreactor length of 33 cm is used, and only the reactor volume fraction with fully developed velocity profiles is analyzed. Empirical surface kinetics for titanium dioxide nanotubes at the photoelectrode surface is considered as a UDF, as discussed in Section 2.1.

The PSTY is defined as the relation between the STY and the electrical power necessary to carry out the photocatalytic process ($P_{ligth}$) per unit photoreactor volume, it is determined as follows,
(6)$$PSTY=\frac{STY}{P_{ligth}/V_{reactor}}.$$

Furthermore, the external effectiveness factor ($E_{ex}$) is calculated as,
(7)$$E_{ex}=\frac{k_{eff}}{k_{reaction}},$$

This factor determines if the global process is limited by the mass transfer rate (i.e., when $E_{ex}\approx 0$) or by the surface chemical kinetics (i.e., $E_{ex}\approx 1$), which depends on the effective reaction rate ($k_{eff})$ defined as,
(8)$$k_{eff}=\frac{k_{reaction}k_{m}}{k_{reaction}+k_{m}}.$$
where $k_{reaction}$ is the first-order reaction rate and $k_{m}$ is the mass transfer rate coefficient.

#### 2.3.3. Photoreactor Length

The minimum hydrodynamic length, which refers to the length at least 70% of the photoreactor volume is reached with developed velocity profiles, is determined. This value was selected considering the study carried out by Jaramillo-Gutierrez et al. [27], in which achieve more than 75% of the reactor volume with fully developed velocity profiles. The fully developed velocity profiles must be achieved because it leads to good homogeneity of the other phenomena such as mass transport and kinetics throughout the photoreactor [27].

The PM modeling approach to solve momentum transport equations is explained above in Section 2.1.1 and the computational domain used is shown in Figure 6; it is used to reduce the computational cost and be able to perform CFD simulations with a larger photoreactor volume. The procedure for calculating the minimum hydrodynamic length is shown in Figure 7. First, simulations were carried out in the RGB approach, varying the inlet velocity (for the laminar regime). Then, the porous medium coefficients were determined by a second-order polynomial regression taking into account the pressure drop in the three dimensions (axial, circumferential, and radial). Next, a simulation with the PM approach was performed using the initially determined coefficients, then the velocity profiles and pressure drop were compared concerning the results of the RGB approach. Finally, the porous medium coefficients were modified until a correlation coefficient ($R^{2}$) greater than 0.95 was achieved.

Once the porous media model was validated, the length of the photoreactor was varied, and the velocity profiles along the photoreactor were determined every 3 cm. The length of the photoreactor was varied until 70% of the reactor had fully developed velocity profiles.

## 3. Results and Discussion

This section presents the results to select a photoelectrocatalytic reactor flow configuration, its operational variables, and volume. First, the flow configuration of the photoreactor is made by a hydrodynamic characterization considering streamlines and velocity profiles behavior through contour plots and XY plots; likewise, the STY results are presented in XY plots. Second, the operating variables of inlet velocity and energy consumption are determined by maximizing the PSTY. XY graphs are presented with the influence of the hydrodynamic regime (Re) and the intensity of radiation in the STY, PSTY, and $E_{ex}$. Finally, the length of the photoreactor necessary to obtain at least 70% of fully developed flux is determined, and contour plots and XY plots show the velocity profiles along the photoreactor.

### 3.1. Selection of the Photoreactor Flow Configuration

#### 3.1.1. Convergence Study

The computational meshing that generated approximately the same error in the studied configurations was considered to compare the flow results of both photoreactors. For this, a mesh independence study was carried out with Richardson extrapolation as mentioned in Section 2.3.1. All studies obtained a mass imbalance of less than $1\times 10^{-12}$. Figure 8 shows the convergence results of STY in terms of a number of elements in photoelectrode surface for a reference case study with ψ equal to 0.75 cm for both geometries (as mentioned in Section 2.3.1), AF and TF, in laminar and turbulent regime. In addition, a convergent behavior in STY (triangle symbol) towards the value of the STY obtained by Richardson extrapolation (dotted line) was observed.

For the Richardson extrapolation, the last STY values obtained are analyzed, ensuring a ratio between the fine and coarse mesh size (coarse mesh size/fine mesh size) of approximately 1.2 to increase the extrapolation precision. It is observed that a computational mesh approximately of $5.5\times 10^{5}$ elements in the photoelectrode surface present a relative error lesser than 15% for both photoreactor flow configurations in the laminar regime; this mesh is achieved with an element size of 3 × 10${}^{-2}$ cm. In the case of the turbulent regime, a computational mesh with 4.38 $\times 10^{5}$ and $8.5\times 10^{5}$ elements on the photoelectrode surface is needed to present an error lesser than 30%, which corresponds to an element size of $2.5\times 10^{-2}$ and $2.25\times 10^{-2}$ cm for the AF and TF configuration, respectively. The computational mesh representing the same error in both photoreactor flow configurations is used for all calculations in the turbulent regime. Therefore, the same error is maintained for all calculations to compare the results. An error of less than 30% could not be obtained concerning Richardson’s extrapolation for the turbulent regime because it exceeded the computational cost (i.e., simulations for the laminar regime cases took about 4 to 6 h; however, in the case of the turbulent regime, the simulation time was around 12 to 15 h. Meanwhile, simulations using the PM approach were around an hour); nevertheless, it was monitored that all the calculations reached convergence when the residuals of the transport variables were less than $1\times 10^{-5}$ and that the other convergence graphs (mentioned in Section 2.2) were stabilized. It should be noted that this computational mesh size was used for the other studies.

#### 3.1.2. Hydrodynamic Characterization

The hydrodynamic characterization is performed under laminar and turbulent conditions for both photoreactor flow configuration, AF and TF, varying the distance between electrodes ψ (i.e., for values of 0.75, 1, and 1.25 cm) by the RGB modeling approach. For example, Figure 9 shows the streamlines for the AF and TF geometry and a constant ψ of 0.75 cm; it can be seen that the AF geometry mainly develops an axial flow, while the TF geometry develops a helical flow.

The influence of ψ on the velocity profiles for each photoreactor flow configuration is analyzed. The results shown correspond to the laminar regime; however, the calculations were made in both the laminar and turbulent regimes. Thus, results are presented in such a way to visualize the tendency and the analysis that it wants to explain. It should be noted that for the study, all the results were considered.

Figure 10 shows the velocity magnitude contours in a cross-sectional plane at 25 cm from the inlet for both geometries, with a value of ψ of 0.75 cm in the laminar. It was found that decreasing the value of ψ does not represent a significant wall effect between both electrodes. It can be seen in Figure 10 that when ψ is 0.75 cm, a zone of low velocities between the electrodes is not created. This behavior was present in both photoreactor flow configurations and could be due to the presence of the photoelectrode mesh, which generates microturbulence that decreases the viscous effects. In addition, there is greater homogeneity in the flow when the TF geometry is used; it is observed in the velocity magnitude contours of Figure 10 that the AF geometry creates a high-velocity zone. This would probably cause a short-circuit with low residence times in this zone, which would not benefit the photoreactor performance. The same tendency was observed in turbulent regimes.

Figure 11 shows the axial velocity contour for AF and TF configuration along the axial axis and the circumferential velocity profile for both geometries at a constant ψ of 0.75 cm. A hydrodynamic behavior divided into three zones was created as shown by the axial velocity contour in Figure 11. Zone I corresponds to the region between the photoelectrode and the external surface of the photoreactor. Meanwhile, Zone II is the region between the electrodes, and Zone III is between the cathode and the surface of the internal tube of the photoreactor. These three zones’ hydrodynamic behavior was also observed in the turbulent regime, with the maximum velocity in Zone I. In Figure 11, it is also observed that the TF geometry, due to the fluid flow helical motion, presents high velocities in the circumferential direction compared to AF, as shown in the circumferential velocity profile (right side). The highest circumferential velocities were presented for both laminar and turbulent regimes; this explains the behavior of the STY as seen in the Section 3.1.3. In addition, this behavior decreases the axial velocity in Zone I in TF flow configuration, as shown in velocity contours in Figure 11. The maximum axial velocity in this zone in the laminar regime is $5\times 10^{-3}$ m/s for the TF geometry, while for AF is $6\times 10^{-3}$ m/s. This increases the residence time in the TF flow configuration and homogeneity in velocity profiles. In the turbulent regime, the same tendency was observed; the TF flow configuration presented a maximum axial velocity of $10\times 10^{-2}$ m/s in Zone I, while the AF geometry had a maximum axial velocity of $12.5\times 10^{-2}$ m/s. Other authors have studied this effect of the helical flow on the fluid residence time distribution [28], who found that a helical movement increases the contact time of the particles and the fluid inside the reactor, and more intense dynamics of macromixing as a result of more significant velocity gradients and turbulent intensities.

#### 3.1.3. Effect in Space-Time Yield

The effect of ψ on the STY of each photoreactor flow configuration was evaluated. The STY is a factor used to compare photoreactors since it relates to the amount of degraded dye concerning the residence time and the photoreactor volume. Tangential inlet and outlet flow improved the STY for both laminar and turbulent regimes. For example, it is shown in Figure 12a that with the TF flow configuration, the STY can be increased up to 89 g/m^3^-day compared to AF (79 g/m^3^-day) considering the same conditions for both configurations and laminar regime. This is due to fluid helical fluid motion generated by the tangential inlet and outlet, conducing to the presence of velocity in both the axial and circumferential directions, increasing the mixing and mass transfer rate in both directions. At the same time, the AF configuration generates a purely axial flow. The same tendency to increase the STY with TF was observed in the turbulent regime. Figure 12b shows that for ψ equal to 1.25 cm and using a TF configuration, the STY is 176 g/m^3^-day, while with the AF configuration, a value of 130 g/m^3^-day. This trend is related to the increase in the velocity circumferential component due to the helical flow motion in the TF, as was observed for the laminar regime.

The distance between electrodes does not represent a significant effect on the STY as shown in Figure 12. This is due to the photoreactor’s geometric configuration, the electrodes’ arrangement, and the three-dimensional woven mesh photoelectrode’s presence and dimensions, which allow the mass transfer resistance to be reduced at the macroscopic level. Therefore, decreasing ψ does not generate a significant flow resistance to decrease the mass transfer rate and consequently the STY. The TF geometry and a ψ of 0.75 cm are selected; the latter, because it does not generate a significant decrease in the STY and a smaller distance between electrodes, favors the reduction of ohmic losses.

### 3.2. Operating Variables

To determine the operating conditions, a photocatalytic Space-time Yield was maximized by modifying variables such as the inlet velocity and energy consumption of the illumination system.

Initially, the STY was analyzed by varying the inlet velocity at a constant surface radiation intensity of 7 W/m^2^, which is possible to obtain from solar radiation (<12 W/m^2^). It can be seen in Figure 13a that the STY increases up to a value of 30 g/m^3^-day when the inlet velocity is 0.03 m/s ($Re_{in}=424$) at a surface radiation intensity of 7 W/m^2^. It is observed that increasing the inlet velocity above this value does not increase the STY; this is because the reaction rate begins to be significant (i.e., $E_{ex}>0.6$) as shown in Figure 13b; therefore, improving the STY by increasing surface radiation intensity is more reasonable.

At a surface solar radiation intensity of 12 W/m^2^, a STY of 60 g/m^3^-day can be obtained with an inlet velocity of 0.2 m/s ($Re_{in}=2830$); above this velocity, the increase in STY is not significant because the $E_{ex}$ remains greater than 0.6. For that reason, the surface radiation intensity is increased to a value of 80 W/m^2^ so that the $E_{ex}$ is less than 0.2; it is shown that the STY increases significantly up to a value of 158 g/m^3^-day with an inlet velocity of 0.3 m/s; however, increasing the inlet velocity does not generate a significant increase in the STY.

On the other hand, the influence of inlet velocity and surface radiation intensity on PSTY was analyzed as shown in Figure 14. The PSTY, contrary to the STY, presented the maximum value under solar radiation. For a surface radiation intensity of 12 W/m^2^ and an inlet velocity of 0.2 m/s, a PSTY of 45 g/day-kW is obtained by increasing the surface radiation intensity to 80 W/m^2^, and the PSTY decreases to a value of 4.5 g/day-kW because the energy increase is more significant than the STY improvement. Therefore, it is possible to affirm that the PSTY can be maximized up to a value of 45 g/day-kW at an inlet velocity of 0.2 m/s, solar radiation for external illumination, and internal radiation by UV-LEDs of 14 W/m^2^.

The designed photoreactor configuration allows both solar and artificial illumination, improving the illumination performance by keeping the entire photoanode surface illuminated and reducing energy consumption using UV-LEDs as an artificial illumination source. In addition, the photoreactor design procedure based on the maximization of the mass transfer rate and the geometry of the photoelectrode allows for obtaining a high STY, which, together with the illumination system, is represented in high PSTY values. For example, Turolla et al. [39] used a tubular photoelectrocatalytic reactor to degrade the azo dye Direct Green 26; they used a mesh photoelectrode with TiO_2_ nanotubes. However, the photoreactor design procedure was not specified. As a result, they achieved 88% decolorization in 24 h using a light source of 8 W, resulting in a PSTY of 3.5 g/day-kW. Likewise, Li et al. [40] developed a highly efficient rotating disk photoelectrocatalytic reactor to degrade Rhodamine B; nevertheless, the calculated PSTY was 14.5 g/day-kW (the energy consumed by the rotating disk was not taken into account).

#### Photoreactor Length

The minimum hydrodynamic length was calculated to achieve at least 70% of the developed velocity profiles in the laminar regime. For this, CFD simulation was used by the PM approach. The porous media coefficients were determined by the methodology explained in the Section 2.3.3. Figure 15 shows the fully developed velocity magnitude profiles obtained with the RGB and PM approach under the same flow conditions. It is observed that there is a good prediction of the velocity profile with the PM approach once the porous medium coefficients were adjusted, and a correlation coefficient $R^{2}$ of 0.988 was obtained.

Figure 16 shows the axial velocity magnitude contours for the RGB (top side) and PM (lower side) approach with an inlet velocity of 0.03 m/s (laminar regime) and a reactor length of 33 cm. It is observed that the PM approach can obtain a hydrodynamic behavior similar to that of the RGB approach; approximately the same region of undeveloped flow is identified (pink box). With this, and the fully developed velocity profiles discussed above, it is possible to confirm that a good approximation of the PM approach can be achieved in the laminar regime; that is, the PM approach can capture the hydrodynamics of the photoreactor with the advantage of being able to model the photoreactor with a larger volume without significantly increasing the computational cost.

Once the coefficients of the porous medium were adjusted, the reactor length was increased, and the velocity magnitude profiles were obtained along the reactor every 3 cm. The length of the photoreactor was increased until at least 70% of the volume with fully developed velocity profiles was achieved. Figure 17 shows the velocity profiles every 3 cm for a photoreactor with a length of 70 cm; it is observed that after 18 cm, the velocity profiles stabilize, representing approximately 75% of the photoreactor with fully developed velocity profiles. In addition, in the fully developed hydrodynamic profile obtained with the reactor length of 70 cm in the laminar regime, no significant difference in the maximum velocities in Zone I, II, and III are observed. With this, it is possible to obtain equivalent residence times in each zone, increasing the homogeneity level along the photoreactor.

Therefore, a length of 70 cm is sufficient to achieve at least 70% of the photoreactor volume with fully developed velocity profiles in laminar regime. It is necessary to mention that in this work, a study was carried out to determine some crucial factors in the design of a photoelectrocatalytic reactor to degrade dyes, such as the flow configuration of the photoreactor, its operational variables, and the volume of the photoreactor. This study provides a computational approach to obtain conclusions from these factors. To improve the accuracy and reliability of the results, mesh independence studies with Richardson extrapolation were carried out; each of the simulations was monitored until adequate convergence was achieved, as mentioned in the Section 2.2.

## 4. Conclusions

A photoreactor flow configuration, a photoreactor length, and operating variables are selected to maximize the Photocatalytic Space-time Yield in a tubular photoelectrocatalytic reactor for dye degradation. A photoreactor based on tangential flow inlet and outlet was established, producing better space-time yield and a more homogeneous velocity profile than a purely axial flow geometry. Likewise, the distance between electrodes did not significantly affect the space-time yield.

A photoreactor length of 70 cm achieves at least 70% of the volume with fully developed profiles in the laminar regime, which helps to homogenize the mass transport phenomenon inside the photoreactor. The operating variables of inlet velocity and energy consumption of the illumination system were determined considering a photocatalytic space-time yield maximization analysis. It was found that maximizing the PSTY, and it is necessary to use external solar radiation as a source of energy to reduce energy consumption and operate in a turbulent regime. Therefore, under the conditions evaluated, a maximum PSTY of 45 g/day-kW is obtained with an inlet velocity of 0.2 m/s (inlet Reynolds of 2830), external solar radiation, and an internal UV illumination system (such as UV-LEDs) of 14 W/m^2^.

This work proposed a methodology based on a numerical approach to study photoelectrocatalytic reactor operation variables to degrade textile dyes, which can be used with other types of dyes, or a mixture of them.

## Figures and Tables

**Figure 1 nanomaterials-12-03030-f001:**
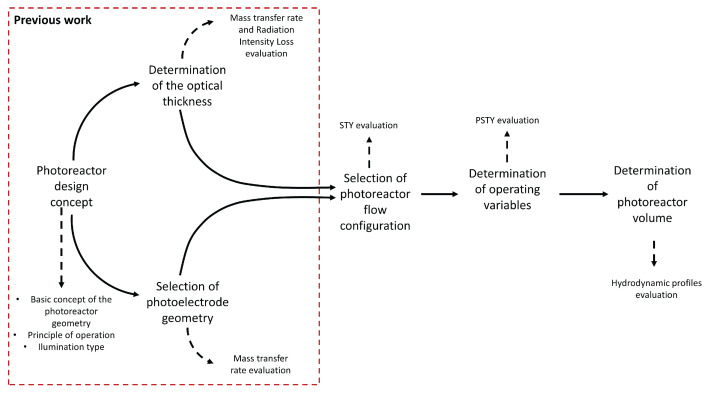
General procedure to design a photoelectrocatalytic reactor.

**Figure 2 nanomaterials-12-03030-f002:**
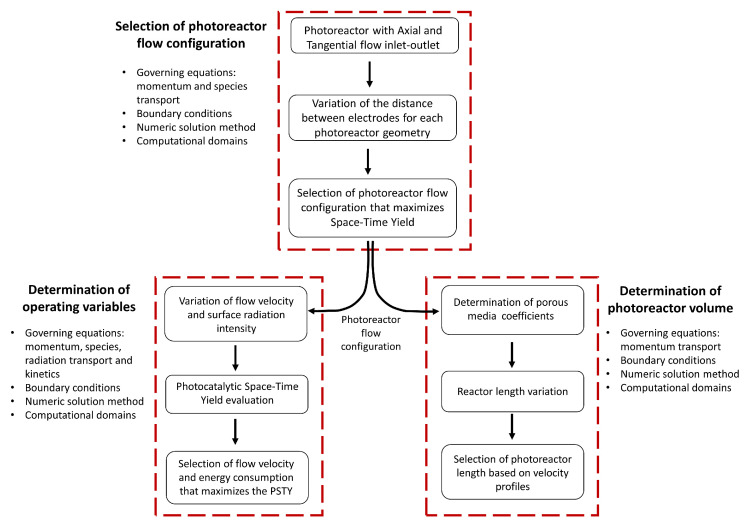
Procedure to select the photoreactor geometry and the operating variables.

**Figure 3 nanomaterials-12-03030-f003:**
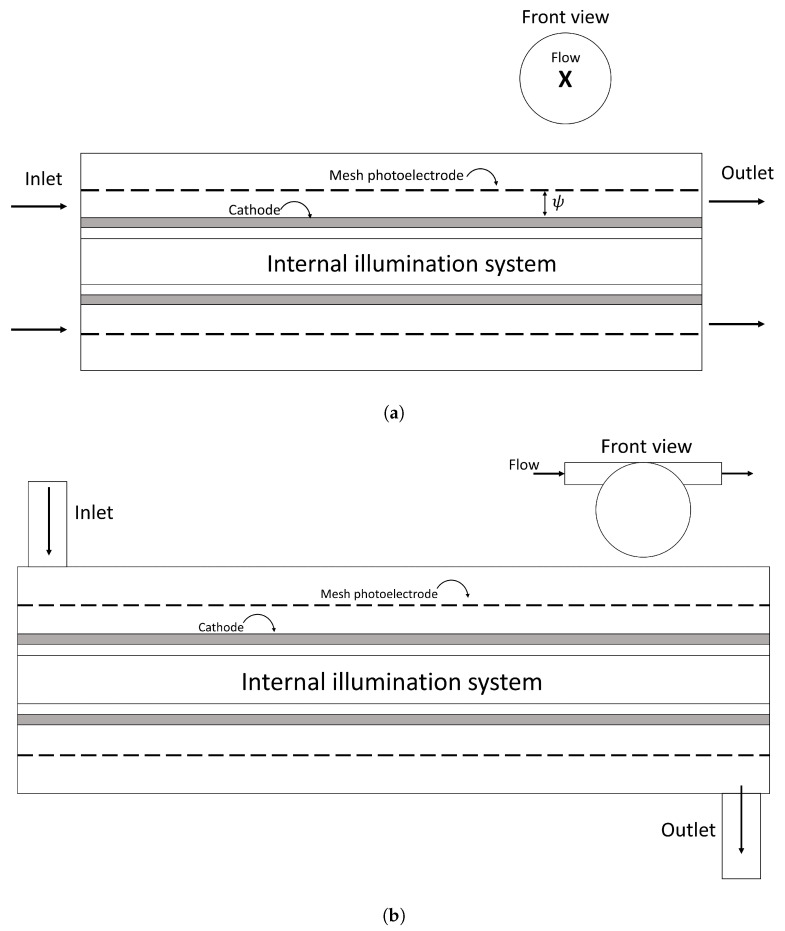
Photoreactor based on (**a**) Axial Flow (AF) and (**b**) Tangential Flow (TF) inlet/outlet.

**Figure 4 nanomaterials-12-03030-f004:**
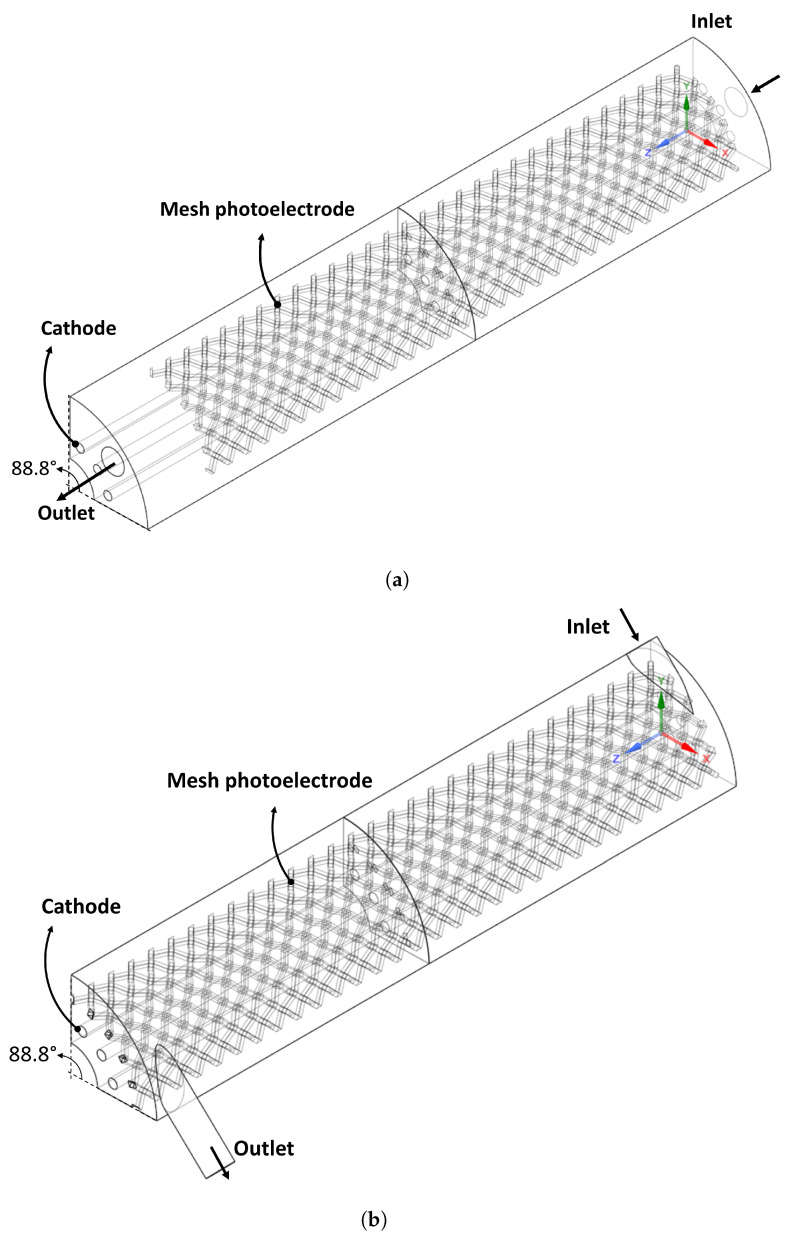
Computational domains used in CFD simulations, (**a**) AF and (**b**) TF.

**Figure 5 nanomaterials-12-03030-f005:**
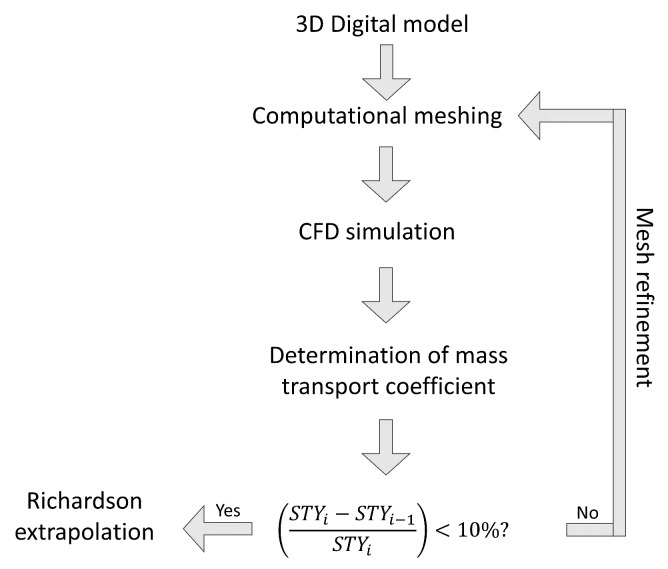
Computational convergence procedure to determine STY.

**Figure 6 nanomaterials-12-03030-f006:**
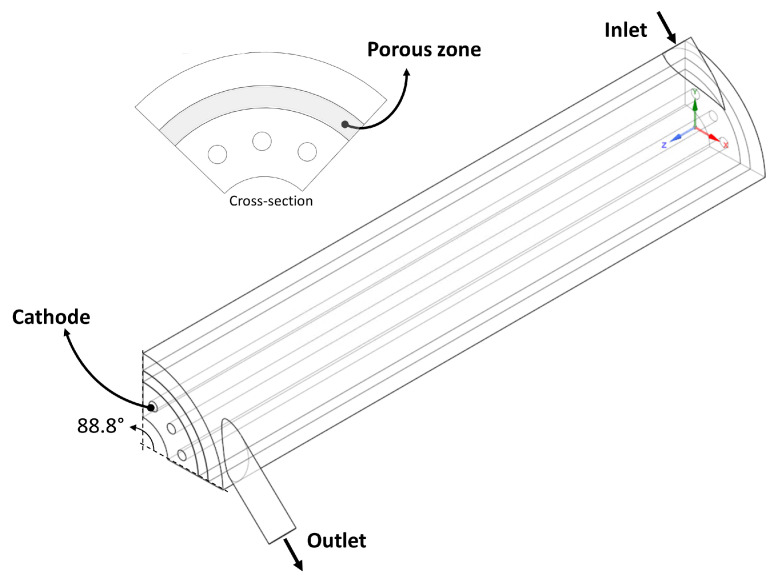
Computational domain used in PM modelling approach.

**Figure 7 nanomaterials-12-03030-f007:**
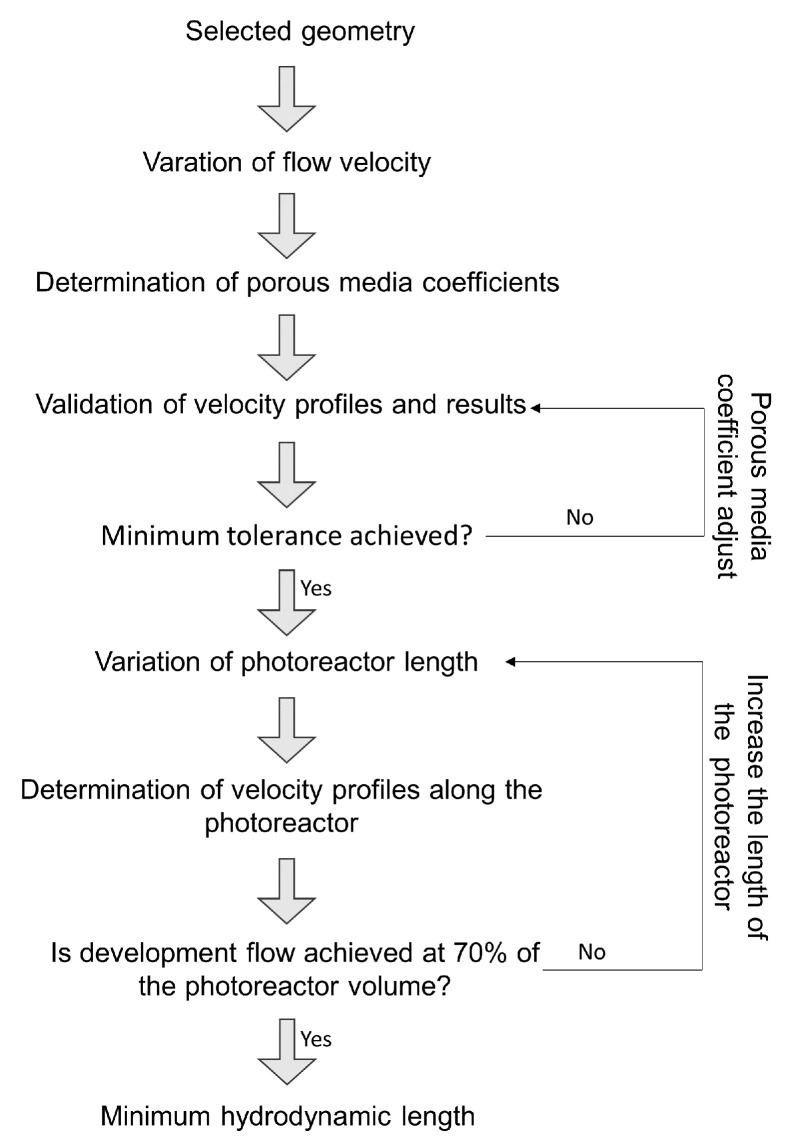
Methodology to calculate the minimum length of the photoreactor.

**Figure 8 nanomaterials-12-03030-f008:**
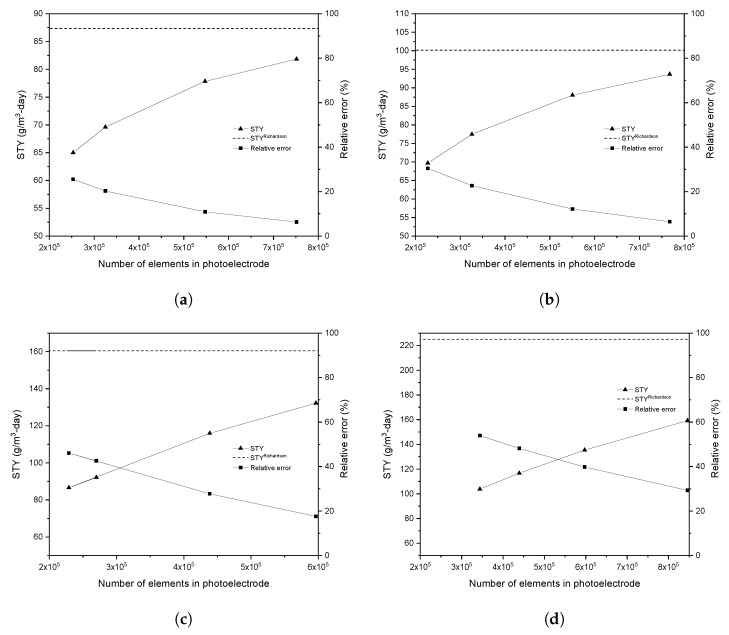
Convergence results for STY for (**a**,**c**) AF and (**b**,**d**) TF configuration in laminar (**a**,**b**) and turbulent (**c**,**d**) regime.

**Figure 9 nanomaterials-12-03030-f009:**
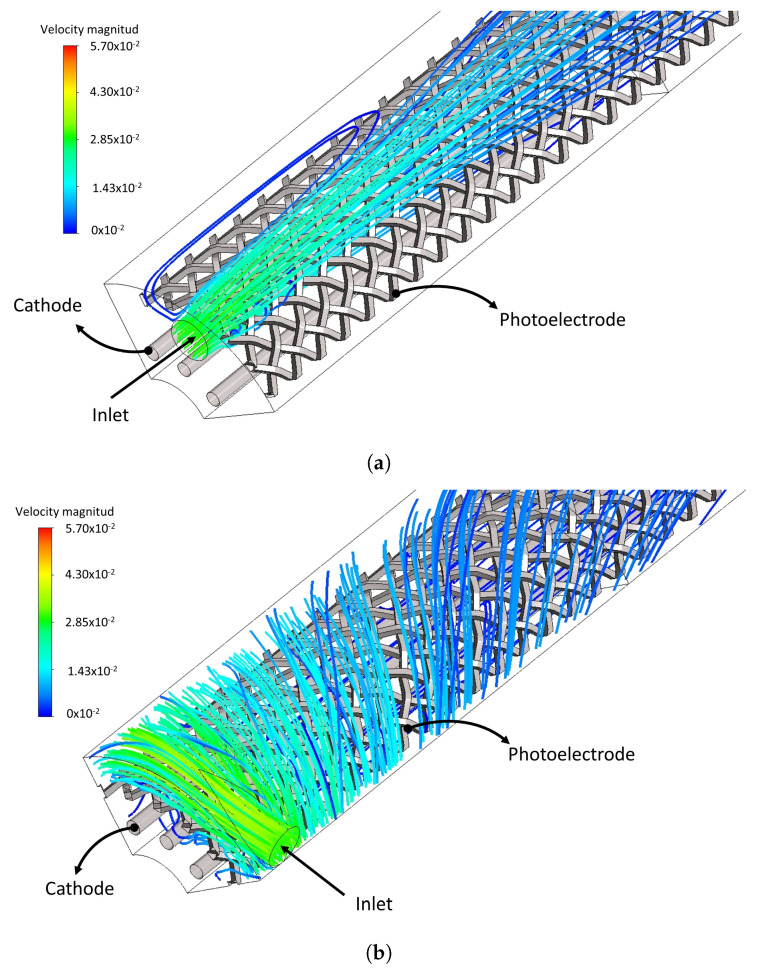
Streamlines for (**a**) AF and (**b**) TF in laminar regime.

**Figure 10 nanomaterials-12-03030-f010:**
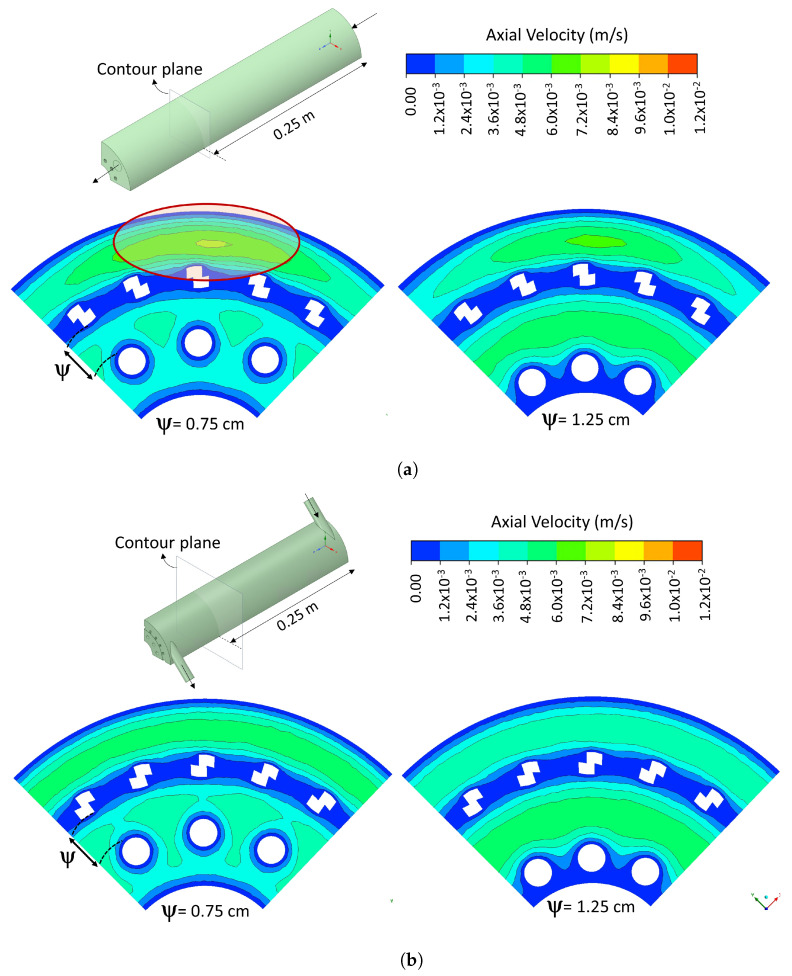
Cross-sectional contour of velocity magnitude with ψ equal to 0.75 and 1.25 cm for (**a**) AF and (**b**) TF configuration in laminar regime.

**Figure 11 nanomaterials-12-03030-f011:**
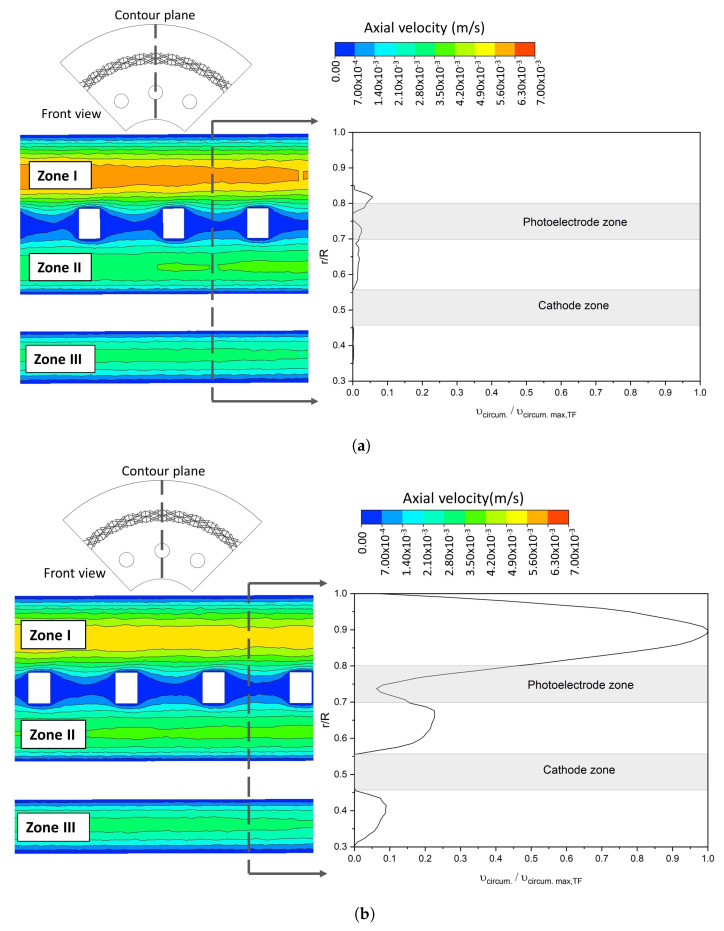
Axial velocity contour along axial axis and circumferential velocity profile at constant ψ (0.75 cm) and inlet velocity of 0.03 m/s for (**a**) AF and (**b**) TF configuration in laminar regime.

**Figure 12 nanomaterials-12-03030-f012:**
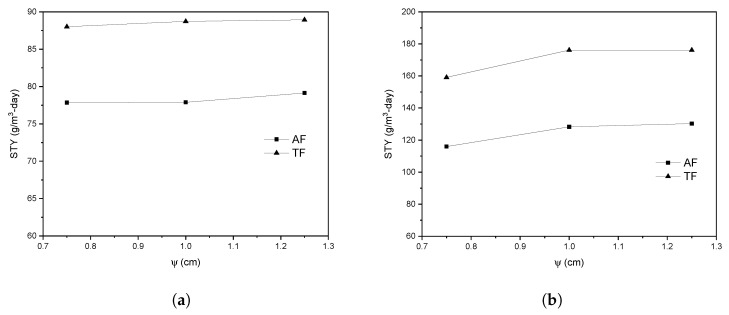
Effect of ψ and photoreactor flow configuration in Space-time Yield in (**a**) laminar and (**b**) turbulent regime.

**Figure 13 nanomaterials-12-03030-f013:**
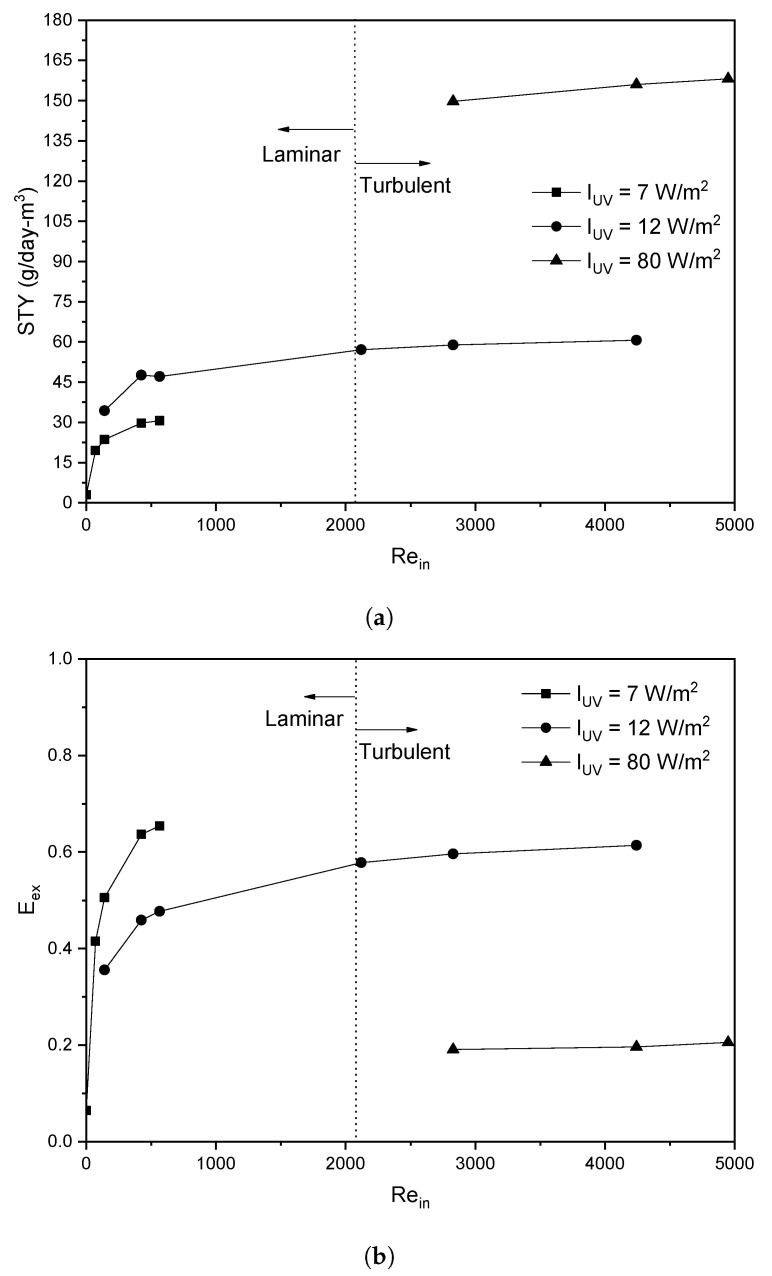
(**a**) Space-time Yield, and (**b**) Effectiveness External factor in terms of Reynolds at the inlet.

**Figure 14 nanomaterials-12-03030-f014:**
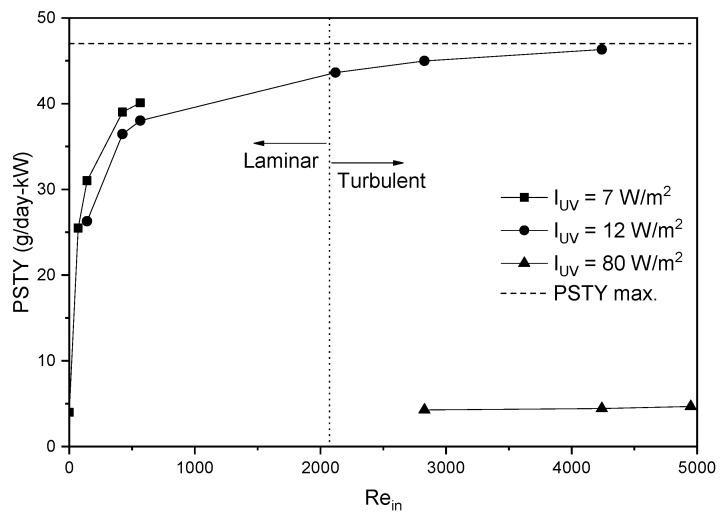
Photocatalytic Space-time Yield maximization.

**Figure 15 nanomaterials-12-03030-f015:**
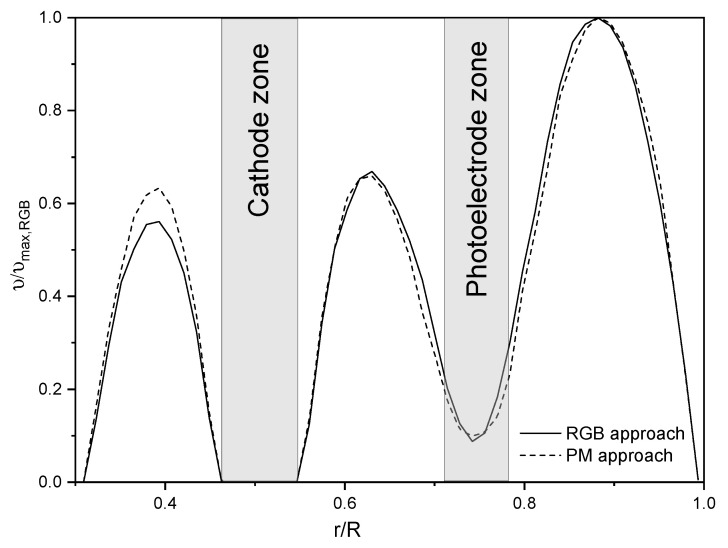
Fully developed velocity magnitude profile using the RGB and PM approach.

**Figure 16 nanomaterials-12-03030-f016:**
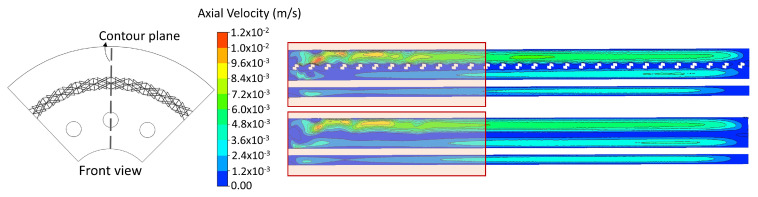
Axial velocity contours using the RGB and PM approach.

**Figure 17 nanomaterials-12-03030-f017:**
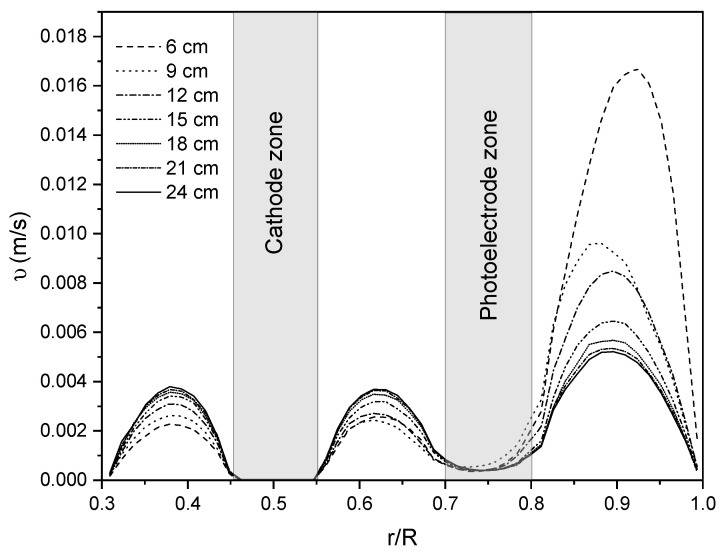
Velocity magnitude profile every 3 cm along photoreactor length (70 cm) using the PM approach.

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
