# Peer review of "Towards the Configuration of a Photoelectrocatalytic Reactor: Part 2—Selecting Photoreactor Flow Configuration and Operating Variables by a Numerical Approach"

_nanomaterials, 2022, doi:10.3390/nano12173030_

Round 1

Reviewer 1 Report

The authors used the numerical method to approach the maximized photocatalytic space-time yield in a photoelectrocatalytic reactor for dye degradation. The axial flow and tangential flow inlet/outlet are designed to evaluate the PTSY. They found that the TF inlet/outlet has a better PSTY. Other variables, such as inlet velocity, external solar radiation, and the power density of an internal UV illumination system are obtained for maximizing the PSTY. The manuscript is well written. It can be published with minor revision.

1. Line 199: What is UVA-average spectrum? The "UVA" acronym is not defined in the text.

2. Line 231: "...a computational mesh with 4.5 × 10^5 and 8.5 × 10^5 elements on the photoelectrode..." In Fig. 8(c) and 8(d), the third X-value is not exactly 4.5 × 10^5. It is a bit less than 4.5 × 10^5 (maybe 4.4 × 10^5).

3. What is the computational cost? Does it have a specific time or ...? Can the authors define it?

Author Response

File with answers is submitted

Reviewer 2 Report

- An overview on supported/immobilized catalysts for photocatalytic oxidation must be given in introduction since several supports and configurations are given in literature;

- Section 2.1.- The models equations, surface kinetics and boundary conditions must be given.

- How was the kinetic equations determined?

- 2.1.2. How was a parameter determined?

- Which were the catalytic materials applied?

- How were the simulation results validated? This must be discussed.

Author Response

File with answers is submitted
